# On Singular Perturbation of Neutron Point Kinetics in the Dynamic Model of a PWR Nuclear Power Plant

**Xiangyi Chen [1] and Asok Ray [1,2,*]** 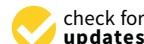

[1] Department of Nuclear Engineering, The Pennsylvania State University, University Park, PA 16802, USA; xxc90@psu.edu

[2] Department of Mechanical Engineering, The Pennsylvania State University, University Park, PA 16802, USA

[*] Correspondence: axr2@psu.edu

**Abstract:** This short communication makes use of the principle of singular perturbation to approximate the ordinary differential equation (ODE) of prompt neutron (in the point kinetics model) as an algebraic equation. This approximation is shown to yield a large gain in computational efficiency without compromising any significant accuracy in the numerical simulation of primary coolant system dynamics in a PWR nuclear power plant. The approximate (i.e., singularly perturbed) model has been validated with a numerical solution of the original set of neutron point-kinetic and thermal–hydraulic equations. Both models use variable-step Runge–Kutta numerical integration.

**Keywords:** PWR nuclear power plants; point kinetics; singular perturbation

## 1. Introduction

The neutron point-kinetics equation with six (or less) delayed groups has been traditionally used for modeling the dynamics of neutron power in nuclear power plants (e.g., pressurized water reactor (PWR)) [1,2]. The neutron point kinetic model is a well-known and experimentally validated model, which is extensively used in both academia and industry. In the integrated (e.g., lumped-parameter) representation of PWR plants, the point-kinetics model is coupled with thermal–hydraulic and fuel heat transfer models to simulate the steady-state and transient behaviors of the reactor and the primary coolant system. This integrated system model, which is represented by (nonlinear) ordinary differential equations (ODEs), is solved numerically because there is no analytical solution, in general. Such numerical simulations are time-consuming especially if the set of ODEs is stiff. Therefore, a challenge in the transient analysis of nuclear plants for design of (real-time) monitoring and (active) control systems is to construct a dynamic model that would be computationally efficient and yet serve the purpose at hand. Since the thermal–hydraulic and fuel heat transfer models act as low-pass filters that attenuate the high-frequency transients of prompt neutrons, the accuracy of predicted transients are not significantly affected if the transients due to prompt neutron are eliminated, provided that net deviations in the total reactivity are small (e.g., in the order of those that may occur during reactor-following/turbine-leading operations of PWR plants [3,4]). A fortiori, when the turbine load changes, due to the thermal capacitance of the pressurized water held in the primary system, which acts a low-pass filter for the secondary system, the states of the primary coolant presents slow dynamics ensuring that there is sufficient time for reactor control.

Over the last several decades, researchers have applied the concept of singular perturbation for numerical analysis of neutron-point-kinetics and thermal–hydraulic dynamics in nuclear power plants (e.g., [1,5–8]). However, in the current state-of-the-art (e.g., [9–13]), different types of nuclear reactors

are still being modeled by including the dynamics of prompt neutrons along with the dynamics of six (or less) delayed groups and temperature reactivity feedback from fuel and coolant.

This short communication addresses the above issue by making use of the principle of singular perturbation [14,15] to approximate the differential equation of prompt neutron as an algebraic equation, which yields a large gain in computational efficiency without compromising any significant accuracy of the simulation results. The main objective here is to demonstrate to practicing engineers, with a rigorous (and yet simple) example, that this approximation of the dynamics of prompt neutrons in the point kinetics model as an algebraic equation indeed significantly improves the speed of dynamic simulation without any noticeable loss of accuracy. This information is very important for control system design in PWR and other types of commercial nuclear power plants.

## 2. Lumped Parameter Model for Simulation

The equations of point kinetics including the dynamics of prompt neutrons and $N_c$ delayed groups ($1 \le N_c \le 6$) are modeled as:

$$\frac{dn(t)}{dt} = \frac{\rho(t) - \beta}{\Lambda} n(t) + \sum_{i=1}^{N_c} \lambda_i c_i(t) \tag{1}$$

$$\frac{dc_i(t)}{dt} = \frac{\beta_i}{\Lambda} n(t) - \lambda_i c_i(t). \ i = 1, \cdots, N_c \le 6 \tag{2}$$

where the terms in Equations (1), (2) and others are defined in the list of abbreviations at the end of this short communication.

At a steady state (ss), having the total reactivity $\rho_{ss} = 0$, $\frac{dn(t)}{dt}|_{ss} = 0$, and $\frac{dc_i(t)}{dt}|_{ss} = 0$, it follows from Equation (2) that:

$$\frac{c_i|_{ss}}{n|_{ss}} = \frac{\beta_i}{\lambda_i \Lambda}, \ i = 1, \cdots, N_c \tag{3}$$

For convenience of analysis, the relative neutron density $n_r(t) \triangleq \frac{n(t)}{n|_{ss}}$ and relative delayed neutron precursors concentration $c_{r,i}(t) \triangleq \frac{c_i(t)}{c_i|_{ss}}$ are obtained by normalization to the fraction of their respective steady-state values. After the normalization, Equations (1) and (2) are rewritten as:

$$\frac{dn_r(t)}{dt} = \left( \frac{\rho(t) - \beta}{\Lambda} \right) n_r(t) + \sum_{i=1}^{N_c} \left( \frac{\beta_i}{\Lambda} \right) c_{r,i}(t) \tag{4}$$

$$\frac{dc_{r,i}(t)}{dt} = \lambda_i n_r(t) - \lambda_i c_{r,i}(t) \text{ for } i = 1, 2, ..., N_c \tag{5}$$

The heat transfer from fuel to coolant and the net heat removal from the coolant are modeled as:

$$P_c(t) = \Omega(T_f(t) - T_c(t)) \tag{6}$$

$$P_e(t) = M(T_l(t) - T_e(t)) \tag{7}$$

The state equations for the lumped fuel temperature $T_f$ and lumped coolant temperature $T_l$ are obtained as:

$$\frac{d}{dt} T_f(t) = (f_f P_a(t) - P_c(t)) / \mu_f \tag{8}$$

$$\frac{d}{dt} T_l(t) = ((1 - f_f) P_a(t) + P_c(t) - P_e(t)) / \mu_c \tag{9}$$

where the average reactor power at time $t$ is obtained in terms of its initial value as:

$$P_a(t) = P_a(0)n_r(t) \tag{10}$$

and the total reactivity due to control rod and temperature feedback from the coolant and fuel is:

$$\rho(t) = \rho_r(t) + \alpha_f(T_f(t) - T_f(0)) + \alpha_c(T_c(t) - T_c(0)) \tag{11}$$

Under a steady-state normal operation, the reactivity is balanced to be zero by the loaded fuel, the control rod position, and the coolant Boron concentration. The coolant and fuel reactivity feedbacks are affected by deviations from the respective temperature reference points. In this short communication, the reference temperatures are set as the initial fuel and average coolant temperatures, respectively; and the control rod reactivity is set as zero before any reactivity insertion. Table 1 lists numerical values of the pertinent parameters and initial conditions, which will allow reproduction of the simulation results presented in this short communication.

**Table 1.** Parameters and Values of parameters used in the simulation [2].

| Parameters | Values [Units] |
|---|---|
| $\Omega$ | 6.53 (MW/°K) |
| $M$ | 92.8 (MW/°K) |
| $\mu_f$ | 26.3 (MW·s/°K) |
| $\mu_c$ | 70.5 (MW·s/°K) |
| $T_e$ | 563.15 (°K) |
| $f_f$ | 0.98 |
| $\alpha_c$ | 0.00001 |
| $\alpha_f$ | −0.00005 |
| $\lambda_1$ | 0.0124 (s$^{-1}$) |
| $\lambda_2$ | 0.0305 (s$^{-1}$) |
| $\lambda_3$ | 0.1110 (s$^{-1}$) |
| $\lambda_4$ | 0.3010 (s$^{-1}$) |
| $\lambda_5$ | 1.1400 (s$^{-1}$) |
| $\lambda_6$ | 3.0100 (s$^{-1}$) |
| $\Lambda$ | 0.0001 (s) |
| $\beta_1$ | 0.000215 |
| $\beta_2$ | 0.001424 |
| $\beta_3$ | 0.001274 |
| $\beta_4$ | 0.002568 |
| $\beta_5$ | 0.000748 |
| $\beta_6$ | 0.000273 |
| $\beta$ | 0.006502 |
| $N_c$ | 6 |
| $n_r(0)$ | 1 |
| $c_{r,i}(0)$ | 1 |
| $T_l(0)$ | 590.09 (°K) |
| $T_f(0)$ | 951.81 (°K) |
| $P_a(0)$ | 2500 (MW) |

Figure 1 shows the dynamics of the reactor parameters after three consecutive steps of control rod reactivity insertion and withdrawal by numerically solving Equations (4)–(11) with Runge–Kutta–Fehlberg method [16], which is an adaptive stepsize version of the standard (fixed stepsize) Runge–Kutta method; the variable stepsize reduces the local truncation error to the order of five. It is noted that the reactivity insertion rate is 3.07 cents per second, which is ∼0.0002 reactivity per second. The simulation code, written in Python 3, takes ∼ 8 s to run on a PC with 4th generation 2.2 GHz Intel Core i7 CPU and 16 GB 1600 MHz DDR3 memory; there are 19,245 steps

with the tolerance value of $1 \times 10^{-6}$ for the estimated local truncation error. The complete code of this paper is uploaded to github, the url is provied at the Supplementary Materials.

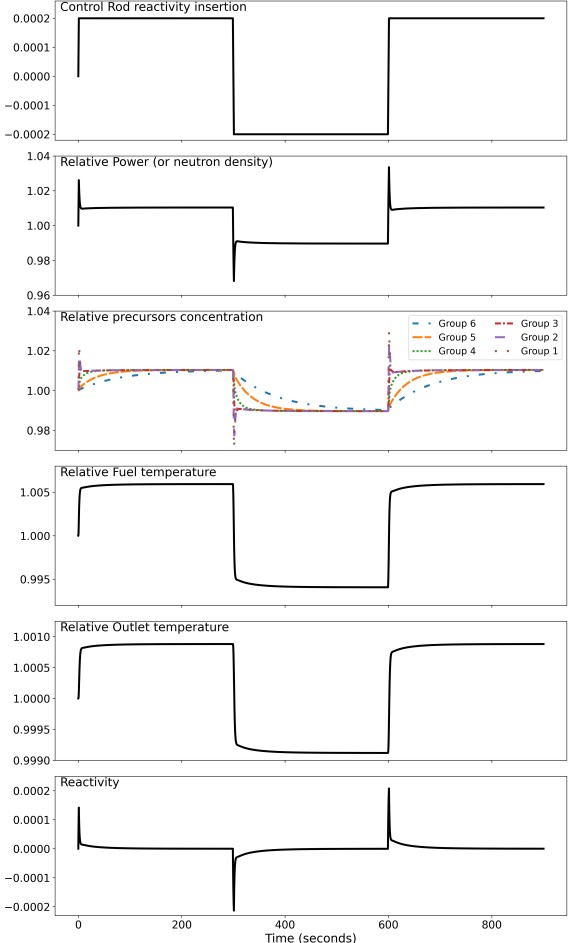

**Figure 1.** Profiles of relative neutron density $n_r$, delayed neutron precursor concentration $c_{r,i}, i = 1, \cdots, 6$, relative average fuel temperature $T_{f,r}$ and relative outlet coolant temperature $T_{l,r}$ with integration step size of 1 ms and 0.9 million steps.

## 3. Enhacement of Numerical Efficiency by Singularly Perturbed Point Kinetics

This section shows how the dynamics of prompt neutrons can be approximated with an algebraic equation, generated by singular perturbation [14,15], and eventually, the step size of integration is dramatically increased to enhance computational efficiency. However, before embarking on construction of singularly perturbed neutron point kinetics, it is necessary to provide a succinct mathematical background, which is presented in the next subsection.

### 3.1. Background of Singular Perturbation

While standard perturbation methods are generally applied to differential equations that are smoothly dependent on a small parameter $\varepsilon$, singular perturbations [14,15] are characterized by discontinuous dependence of system properties on $\varepsilon$, as seen below in the system state equations of a full-order model:

$$\frac{dx}{dt} = f(t, x, z, \varepsilon), \quad x(t_0) = \xi(\varepsilon) \tag{12}$$

$$\varepsilon \frac{dz}{dt} = g(t, x, z, \varepsilon), \quad z(t_0) = \eta(\varepsilon) \tag{13}$$

where the functions $\xi$ and $\eta$ in the initial conditions smoothly depend on the parameter $\varepsilon$ and the initial time $t_0 \in (0, t_1]$. If $\varepsilon$ is sufficiently small (i.e., $0 < \varepsilon \ll 1$), then the time scales of Equations (12) and (13) are treated as slow ($t$) and fast ($\tau$), respectively, because $\varepsilon \underbrace{\dfrac{d}{dt}}_{slow} = \underbrace{\dfrac{d}{d(t/\varepsilon)}}_{fast} \equiv \dfrac{d}{d\tau}$ such that $\dfrac{d\tau}{dt} = \dfrac{1}{\varepsilon}$.

**Remark 1.** *If $0 < \varepsilon \ll 1$ in Equation (13), then the transients of z last for a very short time as they have very high frequency contents due to the fact that $\frac{dz}{dt} = g/\varepsilon$. Having $\varepsilon = 0$ in Equation (13) makes the transients of z instantaneous whenever $g \neq 0$.*

Setting the parameter $\varepsilon = 0$, which is called singular perturbation, causes an abrupt and significant change (e.g., reduction in the dimension of the state space) in dynamical behavior of the full-order system, because the fast-time system (13) degenerates to an algebraic (or transcedental) equation, which is assumed to have at least one isolated real root [14,15].

The reduced-order system is obtained by setting $\varepsilon = 0$ as:

$$0 = g(t, \tilde{x}, \tilde{z}, 0) \tag{14}$$

$$\tilde{z}(t) = h(t, \tilde{x}) \tag{15}$$

where $\tilde{x}(t)$ is the solution of the following equation;

$$\frac{d\tilde{x}}{dt} = f(t, \tilde{x}, \tilde{z}, 0), \quad \tilde{x}(t_0) = \xi(0) \triangleq \xi_0 \tag{16}$$

and $\tilde{z}$ represents the quasi-steady-state of $z$ when $x = \tilde{x}$. The above action may give rise to a discontinuity problem that can be circumvented by analysis in multiple time scales, which is the essence of singular perturbation.

**Remark 2.** *The solutions, $\tilde{x}(t)$ and $\tilde{z}(t)$ of the reduced-order system (i.e., $\varepsilon = 0$) in Equations (16) and (15)), respectively, are expected to be different from $x(t)$ and $z(t)$ in the full-order system in Equations (12) and (13)), respectively. While the unperturbed variable z may start from an arbitrary initial value at time $t_0$, the quasi-steady-state $\tilde{z}$ is not allowed to have an arbitrary initial value at any time t. Therefore, $\tilde{z}(t)$ cannot be a uniform approximation of $z(t, \varepsilon)$; however, if the relation $(x(t, \varepsilon) - x(t)) \sim O(\varepsilon)$ is guaranteed to hold uniformly on an interval $[t_0, t_1]$, then it follows that $(z(t, \varepsilon) - \tilde{z}(t)) \sim O(\varepsilon)$ over the same interval, because*

$$(x(t_0, \varepsilon) - \tilde{x}(t_0)) = (\xi(\varepsilon) - \xi_0) \sim O(\varepsilon)$$

*where a function $\theta : \mathbb{R} \to \mathbb{R}$ belongs to the class $O(\varepsilon)$ if $\lim_{\varepsilon \to 0} \frac{|\theta(\varepsilon)|}{|\varepsilon|}$ exists and is equal to a non-negative real number.*

The concept of the so-called "boundary layer" [14,15] is now introduced to investigate the fast transients in terms of exponential stability of the equilibrium point in the state space. With a change of variable for convenience: $y \triangleq z - h(t, x)$, which shifts the quasi-steady-state of $z$ from $\tilde{z}$ to the origin, the full-order model in Equations (12) and (13) are rewritten in the $(x, y)$ space as:

$$\frac{dx}{dt} = f(t, x, y + h(t, x), \varepsilon), \quad x(t_0) = \xi(\varepsilon) \tag{17}$$

$$\varepsilon \frac{dy}{dt} = g(t, x, y + h(t, x), \varepsilon) - \varepsilon \frac{\partial h}{\partial t}$$

$$- \varepsilon \frac{\partial h}{\partial x} f(t, x, y + h(t, x), \varepsilon), \quad y(t_0) = \eta(\varepsilon) - h(t_0, \xi(\varepsilon)) \tag{18}$$

state of Equation (18) is $y = 0$, which is now substituted into Equation (17) to yield the reduced-order model in Equation (16). By setting $\varepsilon = 0$ in the fast time $\left(\tau \triangleq \frac{t - t_0}{\varepsilon}\right)$ scale, Equation (18) reduces to:

$$\frac{dy}{d\tau} = g(t_0, \xi_0, y + h(t_0, \xi_0), 0), \ \ y(0) = \eta(0) - h(t_0, \xi_0) \tag{19}$$

which is called the "boundary layer" model for singular perturbation.

During the "boundary-layer" interval $[t_0, t_1]$, if the error $(z(t_0, \varepsilon) - \tilde{z}(t_0))$ is indeed $O(\varepsilon)$, then the function $z$ should asymptotically approach $\tilde{z}$ provided that the following conditions, laid out in Tikhonev Theorem [14,15], are satisfied.

1.  The functions $f$ and $g$ in Equations (12) and (13), respectively, and their first partial derivatives with respect to $(x, z, \varepsilon)$ and the first partial derivative of $g$ with respect to $t$ are continuous.
2.  Initial conditions $\xi(\varepsilon)$ and $\eta(\varepsilon)$ in Equations (12) and (13), respectively, are smooth functions of $\varepsilon$.
3.  The function $h(t, x)$ in Equation (15) and the Jacobian $[\partial g(t, x, z, 0)/\partial z]$ have continuous first partial derivatives with respect to their arguments.
4.  The reduced-order system in Equation (16) has a unique solution $\tilde{x}(t)$ for $t \in [t_0, t_1]$ within a compact subset of the solution space.
5.  The origin in the state space of Equation (19) is an exponentially stable equilibrium of the boundary-layer system.

### 3.2. Singularly Perturbed Neutron Point Kinetics

This subsection undertakes the task of singular perturbation of Equation (4) and examines the consequences of this approximation. To reveal the (fast-transient) time constant of neutron kinetics, a time-dependent small parameter $\varepsilon$ is defined as:

$$\varepsilon(t) \triangleq -\frac{\Lambda}{\rho(t) - \beta}, \tag{20}$$

so that Equation (4) can be rewritten as:

$$\varepsilon(t)\frac{dn_r(t)}{dt} = -n_r(t) - \sum_{i=1}^{N_c} \left(\frac{\beta_i}{\rho(t) - \beta}\right) c_{r,i}(t). \tag{21}$$

With a step reactivity insertion $\rho_r(0^+) = 2 \times 10^{-4}$ by the control rod, and after a relatively long time, $T$ (~300 s), during which temperature reactivity feedback takes place, the reactor regain steady-state, $\rho(t) = 0 \ \forall t > T$. The values of $\varepsilon(t)$ before the transients, at the beginning of the transients, and long after the transients are calculated using Equation (20) as: 0.0154, 0.0159, and 0.0154 respectively, as shown in Figure 2. This implies that the mean $\bar{\varepsilon}$ of the variable $\varepsilon(t)$ is always a small number and the ratio of standard deviation to mean for $\varepsilon(t)$ is much smaller than 1, which implies that $\varepsilon(t) \approx \bar{\varepsilon} \ \forall t \in [t_0, t_1]$ during the transients. The small value of $\varepsilon(t)$ also indicates that the neutron dynamics has a small time scale. Singular perturbation (i.e., setting $\varepsilon(t) = 0$) approximates the ODE in Equation (21) by an algebraic equation as:

$$\tilde{n}_r(t) \triangleq -\sum_{i=1}^{N_c} \left(\frac{\beta_i}{\rho(t) - \beta}\right) c_{r,i}(t) \tag{22}$$

**Remark 3.** *The approximation in Equation (22) is a consequence of filtering out fast transients from the original dynamical system. Thus, the boundary layer (i.e., details of the fast time-scale behavior) are eliminated as seen in Figure 3. It is also seen that, beyond the boundary layer, the results of approximation are consistent with the original solution.*

The boundary layer model can now be set in terms of the analytical solution under the assumption that the parameters of slow-dynamics system are constant. Solving Equation (21), the following closed-form solution is obtained:

$$n_r(t) \approx \left[1 + \sum_{i=1}^{N_c} \frac{\beta_i}{\rho(t) - \beta} c_{r,i}(t)\right] \exp\left(-\frac{t}{\varepsilon}\right) - \sum_{i=1}^{N_c} \frac{\beta_i}{\rho(t) - \beta} c_{r,i}(t) \tag{23}$$

**Remark 4.** *The closed-form solution in Equation (23) is interpreted as adding a fast decay term to the algebraic Equation (22), which makes up for the missing transients.*

Figure 3 compares the results generated from Equations (4), (22) and (23). It is noted that Equation (23) only works under the step reactivity insertion case. An example of oscillating reactivity insertion, where the decay term may not die out asymptotically, is presented below.

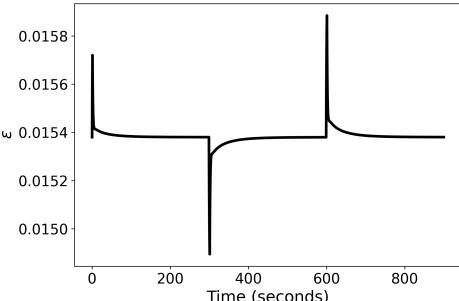

**Figure 2.** Evolution of the singular perturbation parameter $\varepsilon$ with time (Equation (20)).

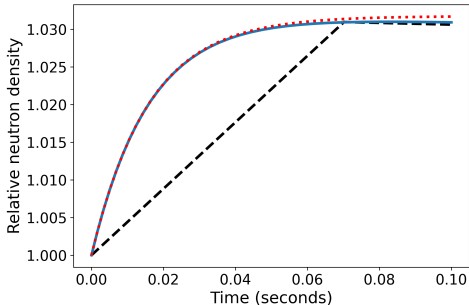

**Figure 3.** Profiles of relative neutron density $n_r$ within the boundary layer, where the solid line —— indicates the profile governed by Equation (4); the dashed line ‑ ‑ ‑ indicates the profile governed by Equation (22); and the dotted line $\cdots\cdots$ indicates the profile governed by Equation (23).

### 3.3. Example: Sinusoidally Oscillating Reactivity Insertion

Let us consider a situation in which the position of the reactor control rod is subjected to small fluctuations around its mean value due to variations in the electrical power generation in the reactor-following/turbine-leading mode of plant operation. This situation is simulated by vibratory motion of the reactor control rod that causes approximately sinusoidal reactivity insertion and withdrawal at 1 Hz frequency with the magnitude of $2 \times 10^{-4}$. To this end, Equations (4) and (22), which represent the original version and singularly perturbed version of prompt neutron model, respectively, are used in the simulation. The computation time and the number of integration steps for the simulation using Equation (4) are 564 milliseconds and 1661, respectively. In contrast, the simulation using Equation (22) takes computation time of $\sim 67$ milliseconds and the number of integration steps is 197. The simulation results are shown in Figure 4, and the relative error $\left(\frac{n_r - \tilde{n}_r}{n_r}\right)$ is plotted in Figure 5. It is calculated that the absolute value of the relative error is bounded by $\sim 0.31\%$, which is considered to be very accurate from the perspectives of numerical simulation. The number of integration steps in

the simulation using the algebraic relation in Equation (22) is approximately one order of magnitude less than that using the ODE in Equation (21) or Equation (4).

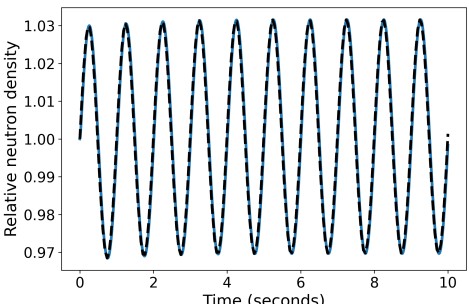

**Figure 4.** Profiles of relative neutron density $n_r$ (Equation (4) simulated with 1661 steps, and indicated by —— ) and $\tilde{n}_r$ (Equation (22) simulated with 197 steps, and indicated by **- - -**) under excitation of sinusoidal oscillations of rod reactivity insertion.

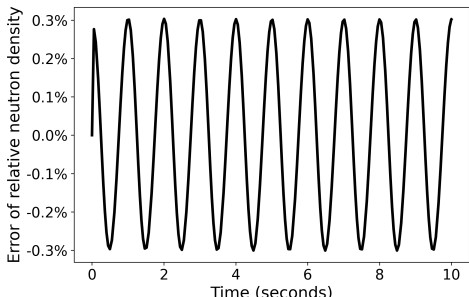

**Figure 5.** Profile of percent error $\left( \frac{(\tilde{n}_r - n_r) \times 100}{n_r} \right)$ for approximation of relative neutron density $n_r$ (see Equation (4) and Equation (22)).

## 4. Discussion and Conclusions

This short communication has demonstrated how the fast dynamics of prompt neutron kinetics can be singularly perturbed to yield an algebraic equation with one order of reduction in the simulation time and without any noticeable loss of accuracy (e.g., peak error in the order of ∼0.31% during transients) and then the error of approximation rapidly approaches zero. Stability of transients is guaranteed if net deviations in the total reactivity are small in the sense that the conditions of Tikhonev Theorem [14,15] are satisfied (see the end of Section 3.1).

Reducing the order of the primary coolant system model by singular perturbation of the reactor prompt kinetics equation is indeed sufficient to investigate its impact on the dynamics of the entire PWR plant. The rationale is that the primary coolant system in a PWR is thermally coupled with the secondary coolant system through the steam generator that, having a reasonably large thermal capacitance, would act as a low-pass filter. Since the primary coolant system practically filters fast transients of prompt neutrons, responses of the secondary coolant system will be even less sensitive to singular perturbation.

**Supplementary Materials:** The code is available at https://github.com/chenxiangyi10/MDPI-sci-On-Singular-Perturbation-of-Neutron-Point-Kinetics-in-the-Dynamic-Model-of-a-PWR.

**Author Contributions:** Conceptualization, A.R.; methodology, A.R., X.C.; software, X.C.; writing–original draft preparation, A.R., X.C.; supervision, A.R.; funding acquisition, A.R. All authors have read and agreed to the published version of the manuscript. These authors contributed equally to this work.

**Funding:** This research was funded by U.S. Air Force Office of Scientific Research (AFOSR) grant number FA9550-15-1-0400 in the area of dynamic data-driven application systems (DDDAS).

**Acknowledgments:** The authors acknowledge the benefits of technical discussion with Shiliang Zhou of North China Electric Power University and William Walters of the Pennsylvania State University.

**Conflicts of Interest:** The authors declare no conflict of interest.

## Abbreviations

The following abbreviations are used in this manuscript:

| | |
|---|---|
| $n$ | neutron density |
| $c_i$ | $i$-th delayed neutron precursor concentration |
| $N_c$ | number of delayed concentration groups $(1 \leq N_c \leq 6)$ |
| $\lambda_i$ | effective precursor decay constant for group $i$ |
| $\Lambda$ | effective prompt neutron lifetime |
| $\varepsilon(t)$ | time-dependent singular perturbation parameter |
| $\bar{\varepsilon}$ | time-averaged singular perturbation parameter |
| $\beta$ | total delayed neutron fraction $(\beta \triangleq \sum_{i=1}^{N_c} \beta_i)$ |
| $\rho$ | reactivity |
| $\rho_r$ | control rod reactivity |
| $P_c$ | power transferred from fuel to coolant |
| $P_e$ | power removed from the coolant |
| $P_a$ | reactor power |
| $\Omega$ | heat transfer coefficient between fuel and coolant |
| $M$ | mass flow rate times heat capacity of coolant water |
| $T_f$ | average fuel temperature in the reactor |
| $T_{f,r}$ | relative average fuel temperature $\left(\frac{T_f}{T_f(0)}\right)$ |
| $T_l$ | coolant temperature at reactor exit |
| $T_{l,r}$ | relative coolant temperature at reactor exit $\left(\frac{T_l}{T_l(0)}\right)$ |
| $T_e$ | coolant temperature at reactor entrance |
| $T_c$ | average coolant temperature in the reactor |
| $f_f$ | fraction of reactor power deposited in the fuel |
| $T_{e0}$ | reference coolant temperature at reactor entrance |
| $T_{c0}$ | reference average coolant temperature |
| $\mu_f$ | total heat capacity of the fuel and structural material |
| $\mu_c$ | total heat capacity of the reactor coolant |
| $\beta_i$ | fraction of neutrons that come from delayed group $i$ |
| $\alpha_c$ | coolant temperature coefficient |
| $\alpha_f$ | fuel temperature coefficient |
| $n_r$ | relative neutron density |
| $c_{r,i}$ | $i$-th delayed neutron precursor's relative concentration |
| $t_0$ | initial time of transients |
| $t_1$ | end time of transients |

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
