# Peer review of "On Singular Perturbation of Neutron Point Kinetics in the Dynamic Model of a PWR Nuclear Power Plant"

_sci, doi:10.3390/sci2020036_

Round 1

Reviewer 1 Report

Overall a really good article which is well presented. 

I feel it might add a bit more depth to the article if the author included a bit more information with regards to the systems codes which use these methods and how this improved method could be implemented. 

The time saving of an order of magnitude is a useful conclusion, however, did the author identify any cases where the time required for these transients were excessive using the standard methods?

Please increase the size of the omega in Figure 2. 

Author Response

I feel it might add a bit more depth to the article if the author included a bit more information with regards to the systems codes which use these methods and how this improved method could be implemented The code of this work is uploaded to github. One can find the code here: https://github.com/chenxiangyi10/MDPI-sci-On-Singular-Perturbation-of-Neutron-Point-Kinetics-in-the-Dynamic-Model-of-a-PWR. In last version, we reported “The computation time and the number of integration steps for the simulation using Equation (4) are 412 milliseconds and 1661, respectively. In contrast, the simulation using Equation (22) takes computation time of ~50 milliseconds and the number of integration steps is 197.” In this new implementation that are uploaded in the github, “The computation time and the number of integration steps for the simulation using Equation (4) are 564 milliseconds and 1661, respectively. In contrast, the simulation using Equation (22) takes computation time of ~67 milliseconds and the number of integration steps is 197.” You can find the ratios of the running time are similar: 8.4 for the old implementation and 8.2 for the new implementation. The manuscript is revised accordingly. The time saving of an order of magnitude is a useful conclusion, however, did the author identify any cases where the time required for these transients were excessive using the standard methods? In the introduction section, it has been addressed that “a challenge in the transient analysis of nuclear plants for design of (real-time) monitoring and (active) control systems is to construct a dynamic model that would be computationally efficient and yet serve the purpose at hand.” For example, in the field of expert system of the nuclear power station, the simulation time reduction by one order makes fundamental difference. It could help the operators/systems identify the type of transient in time. Considering the artificial intelligence are gradually incorporated in the expert system, the system development is unlikely to be carried out by using complied language but by interpreted language like python and matlab because the availability of the supporting packages. However the interpreted language has the issue of low implementation speed which could prevent the efficiency of the expert system. Though the hardware and software are changing time by time that efficiency issues come in and be fixed out. The authors consider the efficiency of the dynamical system alone will find its outlets on its own merit and will not envision the application circumstances in detail in the manuscript. However, it is important to keep in mind about the valid domain that the method can be used. The valid domain of the method requires the small positive ϵ. In the case of point kinetics, ϵ(t)=-Λ/(ρ(t)-β). The accident of “control rod ejection”, for example, causes large reactivity insertion obviously is not under the scope. A small positive ϵ requires ρ(t) is smaller than β and not too close to β that their difference is in the order of Λ. One can monitoring whether the scope is valid when this method is implemented. The example in 3.3 has nothing to do with the load following. We mentioned the load following for explaining the thermal-hydraulic process is slow dynamic and neutron kinetic process is fast dynamic. Please increase the size of the omega in Figure 2. The comment is unclear. The authors did not include omega in Figure 2.

Reviewer 2 Report

The paper is very well written and insightful. It is very beneficial to the community at large.

I have a few minor suggestions that may help readers better understand the potential for this work:

I would encourage the authors to include a discussion about the scope of applicability of their proposed approach. It would be helpful to perhaps list a few example transient scenarios and distinguishing between those where the assumptions hold and ones where it does not (perhaps in a table?). The example regarding prompt reactivity insertion is very helpful, but what about cases beyond this? It is hard to imagine other accident scenarios that fit the 'abrupt and singular perturbation' requirement. The authors alluded to load following in the introduction, these feedbacks are slower than the control rod ejection case considered. The example in 3.3 then adds to the confusion as it is almost alluded that fluctuations in loads cause fluctuation in the control rods, which should be an independent system. Perhaps some minor rephrasing here could help.

On a similar note, it is unclear what is meant by the term 'reduction in the dimension of the state space' in page 5. Is this a sudden change in the core geometry somehow?

Lastly, some thoughts on how this approach would apply to reactors beyond PWRs could be helpful. For instance, a fast reactor would have a smaller beta-eff, would that impact the applicability of the novel approach?

Author Response

I would encourage the authors to include a discussion about the scope of applicability of their proposed approach. It would be helpful to perhaps list a few example transient scenarios and distinguishing between those where the assumptions hold and ones where it does not (perhaps in a table?). The theoretical valid domain of the singular perturbation of in the point kinetics is discussed in the section 3. A full scope nonlinear dynamical simulation code of PWR is under development. A plenty of scenarios will be implemented with and without singular perturbation in the future work. The example regarding prompt reactivity insertion is very helpful, but what about cases beyond this? It is hard to imagine other accident scenarios that fit the 'abrupt and singular perturbation' requirement. The authors alluded to load following in the introduction, these feedbacks are slower than the control rod ejection case considered. The example in 3.3 then adds to the confusion as it is almost alluded that fluctuations in loads cause fluctuation in the control rods, which should be an independent system. Perhaps some minor rephrasing here could help. The valid domain of the method requires the small positive ϵ. In the case of point kinetics, ϵ(t)=-Λ/(ρ(t)-β). The accident of “control rod ejection”, for example, causes large reactivity insertion obviously is not under the scope. A small positive ϵ requires ρ(t) is smaller than β and not too close to β that their difference is in the order of Λ. One can monitoring whether the scope is valid when this method is implemented. The example in 3.3 has nothing to do with the load following. We mentioned the load following for explaining the thermal-hydraulic process is slow dynamic and neutron kinetic process is fast dynamic. On a similar note, it is unclear what is meant by the term 'reduction in the dimension of the state space' in page 5. Is this a sudden change in the core geometry somehow? The dimension of the state space means the number of state variables in the ODEs. Lastly, some thoughts on how this approach would apply to reactors beyond PWRs could be helpful. For instance, a fast reactor would have a smaller beta-eff, would that impact the applicability of the novel approach? Yes, it could impact the applicability. See the response to the comment 2.

Round 2

Reviewer 1 Report

na

Reviewer 2 Report

The authors clarified my comments/suggestions.